# Motivations of Young Women Volunteers during COVID-19: A Qualitative Inquiry in Bahrain

**Debashish Sengupta [1,\*] and Dwa Al-Khalifa [2]**

[1] College of Business and Financial Sciences, Royal University for Women, West Riffa 942, Bahrain
[2] Center for General Studies, Royal University for Women, West Riffa 942, Bahrain; dalkhalifa@ruw.edu.bh
\* Correspondence: dsengupta@ruw.edu.bh

**Abstract:** Volunteering work has played a major role in dealing with the COVID-19 pandemic crisis. Studying volunteering behavior is interesting because it holds many important lessons for businesses to attract and engage their primary stakeholders (employees and customers) and counter the challenges posed by the pandemic. As women make up a large percentage of volunteers, understanding the relationship between motivation and women intending to take up volunteering work during crises is necessary—particularly in collectivist Islamic societies. The present study examined the motivations of young women in Bahrain to volunteer for RT-PCR testing and vaccination drives sponsored by the government during the pandemic. The study also examined the effect of the volunteering experience on the lives of these women. The study was conducted using a mixed qualitative method that included focus groups and in-depth interviews. The research participants were millennial women who had undertaken volunteering during the pandemic. A few in-depth interviews were conducted with male volunteers to examine whether such motivations were influenced by gender. The findings of the research revealed normative, in addition to personal, motivators behind the act of volunteering, with a greater dominance of normative motivations such as the call of the homeland and philanthropy. The influence of the collectivist culture in shaping the normative motivations behind volunteering among these women was visible, and there was also an influence of religion and religious values.

**Keywords:** volunteering; women; millennial; pandemic; culture

## 1. Introduction

Volunteering is the act of an individual who contributes their time, effort and resources to help a fellow human being (Gaston and Alexander 2001). It is an act that is performed voluntarily (without any compulsion, by the willful free choice of the individual), without any expectation of payment or profit (although the organization for which the individual volunteers is expected to cover all costs related to the person's volunteering work) (Buric 2004); volunteering must be mutually beneficial (however, the benefits do not include any form of material benefit) (Mihajlovic et al. 2010). Volunteering not only benefits the organizations and communities for which people volunteer, but also benefits the volunteers themselves (Surujlal and Dhurup 2008). It also helps to build connected and stable communities in society by enhancing trust, unity and mutuality among them (Dawkins 2006). Motivation plays a key role in people taking up volunteering work that offers them no monetary benefit, despite the devotion of their time and effort, the training undergone and the potential continuation of their volunteering work (Mertins and Walter 2021). Motivation provides people with a cause to take up volunteering work, drives their intention to sustain it and explains their purpose for volunteering (Ariely et al. 2009). Volunteering motivation can be classified into utilitarian, affective and normative incentives. Utilitarian benefits are the indirect benefits that one gains from volunteering in terms of new knowledge and skills (Davies 2017). Affective benefits are associated with better or improved interpersonal relationships because of positive social interactions during volunteering. Volunteers

may experience an enhanced sense of prestige, self-esteem, engagement, respect for self and others and friendship with other volunteers, which have a positive impact on their interpersonal relationships (Ji et al. 2020). Normative incentives are in the form of a desire to promote the welfare of others (philanthropy) and showing a selfless concern for the wellbeing of others (altruistic) (Aranda et al. 2019).

Millennials are not only one of the most socially minded generations, but they tend to volunteer more for causes than for organizations (The Millennial Impact Project 2013). They prefer to volunteer for causes that appeal to them. However, at the same time, gender differences in motivations behind volunteering have also been established (Burns et al. 2008). Women are strongly connected to their communities and have a desire to give back to society. Despite the fact that they find themselves short of time owing to family duties and responsibilities, they seek volunteering opportunities. A study by the OECD showed that women were at the core of the fight against the pandemic (OECD 2020). In Bahrain, women led this fight through an active participation in leadership and volunteering efforts (SCW 2020).

The objective of the present paper is to contribute to an understanding of the motivation of young millennial women volunteering during the COVID-19 pandemic in Bahrain in a collective context, where the personal is happily sacrificed on the altar of collective good and community welfare. The study also helps to describe how the volunteering experience influenced the lives of the female volunteers. Bahrain was chosen for the study for a variety of reasons—it emerged as one of the top nations in effectively combating the pandemic; almost half of the nation's population is constituted of women, and they also represent nearly half of the country's workforce. This is further explained later in the paper in the context section. Considering that both authors are based in Bahrain, they had a ringside view of the volunteering activity during this crisis. Further, since no such studies have previously been conducted, this also lends a scientific novelty to the findings. The qualitative study was conducted on millennial women in Bahrain, aged between 22 and 28 years. The role and influence of culture, generation and gender in influencing their motivations was also examined. The findings help to illustrate what motivates young women to volunteer, which, in turn, may be helpful for governments and non-profits to mobilize this important cohort of volunteers, especially during emergencies and crises. The findings also indicate that young women are highly socially minded and socially driven in this nation, and this explains why companies should undertake efforts to attract and retain this significant talent base. For businesses, understanding the motivations of these socially driven women may also be important to devise strategies to attract and engage this sizeable customer base.

## 2. Literature Review

### 2.1. Volunteering

Volunteering is defined as any action in which one individual, team or organization willingly gives their time to help another person, group or institution. It is debatable whether these advantages can include monetary compensation. Some scholars claim that remunerated labor is not genuinely voluntary (Smith 1994), while others believe that those who want to work in low-paying occupations, with an objective of assisting others, should be regarded as "quasi-volunteers" (Smith 1982). The inclusion of some relation to intentions in the concept of volunteering is also subject to dispute. Some people believe that volunteering is defined by a desire to serve others. Others argue that volunteering entails working for a "public" good, without regard to motivation (Clary and Snyder 1999). This behavioral approach is consistent with the new perspective on volunteering as a productive activity, because volunteering is defined as creating products and services at a lower-than-market rate, with no consideration of the motives for participation (Wilson 2000).

Volunteering is one of a larger group of actions intended to assist others. Volunteerism is often proactive rather than reactive, unlike the impulsive assistance offered to an attack

victim, where it is important to determine whether or not to intervene quickly, and the contact is short and frequently disorganized. It necessitates a time and effort investment. Caring is commonly connected to one's emotional work on behalf of family and friends, whereas volunteering is more organized and public (Snyder and Omoto 1992). There are a few clear distinctions between these activities: the care connection implies a level of work that is absent from volunteering. Most of the social engagement that is correctly termed volunteering is merely caring in the broadest sense.

### 2.2. Prominent Features of Volunteering

Volunteering is not an objective in and of itself; instead, it is an action that positively impacts the community. In the conventional perspective on volunteering, it is assumed that it is motivated only by compassion. According to a more modern perspective, people volunteer to create a beneficial outcome for both society and themselves (Wilson 2012).

Volunteer labor is clearly not paid labor, but the associated statement in the Volunteer Rights declaration that contributors should not be out of budget due to their volunteering activities might cause some misunderstanding (Smith 1994). Volunteers may still be paid fixed fees or charitable donations based on the amount spent on their voluntary work. When volunteering abroad, they may be compensated for expenditures made on behalf of the organization for which they serve or receive a small living allowance.

Volunteering is distinguished from civic responsibilities and other unpaid work by its freedom of choice. Volunteering relies on the active participation of individuals who appreciate the chance to be involved or via non-profit organizations that serve the community (Snyder et al. 2004).

Regardless of socioeconomic class, cultural origin, education, gender or age, every person has the right to speak up and contribute to their community (Derryberry et al. 2009). Volunteering allows a wide variety of individuals to express this right by providing them with chances to participate with and influence their community, even if they do not choose or are unable to use more official routes of participation or advocacy (Smith et al. 2010).

Volunteering serves a larger societal function than simply providing services, and it is critical that it not be utilized to offer cheap employment or prop up collapsing social structures. Individual volunteers' labor in providing services to others is a means to promote equality, not an aim in itself. Volunteering values are incompatible with involvement in groups (such as racist or white supremacist organizations) that seek to divide the community or prejudice or exclude other particular groups (Derryberry et al. 2009).

Research shows that the relations between social motivations and satisfaction, between social motivations and the emotional exhaustion of volunteers and between growth motivation and satisfaction and the emotional exhaustion of volunteers are moderated by the age of the volunteers (Aranda et al. 2019). Altruistic factors were found to be a motivator when people volunteered to help those with disabilities, and they considered "doing good" as their greatest reward. The research interestingly found that people refer to endogenous motivations when describing their own motivations towards volunteering, while they refer to more exogenous motivations when encouraging others to take up volunteering (Janus and Misiorek 2019). Positive volunteering experiences, such as a desire for interest, social connection, growth of self, recognition and support, have also been found to be drive volunteerism (Same et al. 2020). Volunteering behavior and the mediating role of voluntary motivation were directly and indirectly affected, respectively, by the psychological capital of the volunteers (Xu et al. 2021).

### 2.3. Volunteering during a Crisis

Volunteers play an important role in emergency circumstances (natural disasters or man-made crises such as conflicts or war), attempting to minimize the harm caused by the occurrence (Fernandez et al. 2006). A huge percentage of volunteers have organized other volunteers who enter the volunteering arena via volunteer groups in such situations. Others are self-motivated volunteers who show up on their own. Unlike organized volunteers,

who are prepared for serving in ordinary settings, spontaneous volunteers have not been given any previous training for emergency scenarios and are only trained while working in the field (Whitehead and Smith 2013).

Individuals' contributions in emergency and catastrophe management have been well recorded in disaster studies. The common belief is that disasters create disruption and disarray, with populations becoming helpless, panicked or engaged in antisocial activities such as looting. Individuals and groups, on the other hand, have been observed to become more connected than in "normal" times, frequently cooperating to tackle disaster-related obstacles (Scanlon et al. 2014). In particular, a sociological study has shed light on collective behavior and organizational reactions to emergencies and disasters.

During the current pandemic, three types of volunteering were identified—formal volunteering, social action volunteering and neighborhood volunteering (Mak and Fancourt 2021). People who were more connected with their communities displayed greater intentions to volunteer during the COVID-19 pandemic (Wang et al. 2021). Community identification was found to be positively related to volunteering during COVID-19 and indirectly related to volunteers providing pandemic-related emotional support (Vignoles et al. 2021; Stevenson et al. 2021). Recent research to ascertain the satisfaction with informal volunteering during the COVID-19 crisis on a Swiss online volunteering platform found that volunteering satisfaction during the COVID-19 pandemic was driven by the fulfilment of distinct volunteer motives and platform support (Trautwein et al. 2020).

*2.4. Volunteering and Millennials*

Millennials, unlike previous generations, who sought advancement, authority and jurisdiction, place a greater emphasis on association and success. Millennials are particularly keen to join tribes—communities of people who share the same experiences and ideals—frequently via volunteer work. Social networks play an important role in the communities of millennials and have been proven to be more effective tribal motivators than real-world encounters (Why Tribes Are the New Segmentation 2022). The majority of millennials learn about issues and groups through social networks and communicate with them via Twitter, Facebook or e-mail on their cell phones. Micro-volunteering has become a popular way for millennials to become involved. It enables people to participate in small, simple, crowdsourced ways that only require a few minutes of their time. Micro-volunteering is most typically done through social media posting (e.g., "liking" posts, re-tweeting messages). Millennials engaging at this level are more likely to engage in one-time volunteer activities than group volunteering, and finally leadership roles (Millennial Impact Report 2017), despite being at the end of the volunteer continuum. Micro-volunteering is well-suited to millennials, who face greater challenges in balancing work and family life, compared with previous generations (Twaronite 2015). An empirical study by McLean (2018) indicated that millennials are more compelled than other generations to search for causes such as education, healthcare and the environment as they often feel that they resonate with their ideas.

*2.5. Volunteering and Women*

Globally, women take on most volunteering tasks, at approximately 58% (Krause and Rainville 2018). However, according to a recent empirical study by Gray and Stevenson (2020), their share in volunteering has increased, especially when looking at informal volunteering only. Since volunteering is accomplished by and between individuals, there are significant differences in the ways in which men and women engage in volunteering activities. It often varies in the amount of time spent, type of work done and levels of responsibility between individuals of different genders (Krause and Rainville 2018). However, despite these arguments, undisputable research studies show that females are always more likely to volunteer in social and health services than men, especially in the provision of unpaid care tasks beyond the household (Titus et al. 2017).

On the other hand, men are more likely to volunteer in political, economic and scientific fields than women. According to Gray and Stevenson (2020), the underlying differences are not natural occurrences or rather due to a lack of effort from societies concerned with volunteering activities. In contrast, they could be mirroring existing social norms and structural inequalities in most communities worldwide. Therefore, to achieve a gender balance in volunteering, individuals require more extensive, systemic changes rather than concentrating on encouraging more women to volunteer (Krause and Rainville 2018).

To understand how volunteers might assist communities in becoming more resilient, it is necessary to consider how different groups within a community view volunteerism and resilience. Applying a gender perspective to volunteering for social sustainability is vital, especially when women volunteer more than men across the world (57% and 43%, respectively) (SWVR 2018).

### 2.6. Benefits of Volunteering

One of the more well-known advantages of volunteering is the positive impact that it has on the community. Volunteers who are not compensated are frequently the connection that ties a society together. Volunteering helps to connect with the community while also helping to improve it (Dekker and Halman 2003). Even modest acts of kindness may make a significant difference in the lives of individuals, wildlife and organizations in need.

Committing to a common activity together is one of the most acceptable ways to meet new people and enhance current friendships. Volunteering also improves links to the community and expands the support network by connecting others with similar interests, facilitating access to local resources and providing enjoyable and meaningful activities.

Some individuals are naturally extroverted, while others are introverted and find it challenging to meet new people. Volunteering means practicing and building social skills by allowing one to meet others with similar interests regularly.

Volunteer service engagement has been linked to improved mental and physical health (McDougle et al. 2014), self-esteem (Morrow-Howell et al. 2003), life satisfaction (Thoits and Hewitt 2001), decreased depressive symptoms (Morrow-Howell et al. 2003), happiness (Musick and Wilson 2003), psychological discomfort (Thoits and Hewitt 2001), mortality and functional incapacity (Konrath et al. 2012). Volunteering's medical benefits are not due to self-selection favoritism. Recent longitudinal research found no evidence of reverse causation, wherein volunteering was associated with improved health in the long run but not the other way around. What is yet unknown about the link between volunteering and health outcomes is whether volunteering has cumulative effects on health and what type of volunteering is best for developing volunteer health benefits (McDougle et al. 2014). In terms of the relationship between volunteering and health, the role accumulation approach supports the idea that a volunteer who participates in a variety of volunteer activities at the same time will improve his or her health the most (Morrow-Howell et al. 2003). According to the explanation, role-related social advantages, resources, supporting networks, coping skills, life purpose and gratitude amassed via several prosocial roles can be directly favorable to diverse health outcomes.

### 3. Context: Bahrain/Women

Bahrain has emerged as one of the top nations in the COVID-19 Recovery Index and has won international acclaim for its efforts in dealing with the pandemic. As per the Nikkei COVID-19 Recovery Index, Bahrain ranked number one in the world. The Nikkei COVID-19 Recovery Index assesses the pandemic response of 121 nations based on nine factors, divided into three categories—infection management, vaccine rollouts and mobility (GDN 2021). Bahrain has one of the highest vaccination rates in the world, with 69.81% of its population fully vaccinated, as of 17 March 2022 (Covidvax.Live 2022; CNN 2021). As a country with a population of 1,776,229, Bahrain has, to date, conducted 9,534,009 COVID-19 PCR tests, making it the nation with one of the highest numbers of tests per 1

million population (5,289,634 tests per million population, as of March 2022) (Worldometers 2021).

The phenomenal success of Bahrain in its response to the COVID-19 pandemic and its relentless efforts to safeguard the health and safety of its nationals and residents has been praised by the World Health Organization (Saudi Gazette Report 2021). The country's remarkable success can be attributed to its smart strategies in tackling the pandemic. One such example is when the National Medical Taskforce, tasked with combatting the Coronavirus (COVID-19) pandemic, adopted a COVID-19 Alert Level Traffic Light System. The four alert levels of traffic light level—Green, Yellow, Orange and Red—are based on the positivity rate across various sectors, which then determine the measures accordingly (BNA 2021a, 2021c). The ambitious plans of Bahrain in terms of contact testing and mass vaccination rollouts were met with success largely because of the smart strategies adopted by the government and thanks to the extraordinary team of volunteers, who supported these initiatives through their active participation. The government's invitation to volunteer to support efforts to combat the pandemic (MoH Bahrain n.d.) and the national volunteering platform (NVP, Kingdom of Bahrain n.d.) initiatives to recruit volunteers were met with an excellent response. In a nation with only approximately 1.7 million inhabitants, it was able to recruit more than 30,000 volunteers in a very short period (BTM 2020). Those who joined as volunteers came from both medical as well as non-medical backgrounds.

In Bahrain, 49.4% of all citizens are females, constituting 38% of the total population (Ministry of Information Affairs 2018). The changing role of women is amply evident in Bahraini society. Traditionally, women in Bahrain were restricted to limited roles and were segregated from men (Alsaqer 2018). However, over time, the nation has come a long way and the establishment of the Supreme Council for Women is considered a watershed moment in the quest to empower Bahraini women. Research indicates that "A multiple actor approach where the government takes active ownership for gender equality and development coupled with a decentralized approach to decision-making that favors local (women) based governance was shown to lead to transformational change regarding gender relations in a highly gendered society" (Venugopalan et al. 2021, p. 16). This is, to a large extent, visible in the entire Middle East and North Africa (MENA) region; research shows that the number of women-owned businesses, for instance, has grown significantly (Bastian 2017) and the motivation of women entrepreneurs influenced by various cultural, religious and organizational factors has played a major role in establishing their participation (Bastian et al. 2018). The Supreme Council of Women was established in 2001 and is directly affiliated with His Majesty the King. The establishment of the council has led to the empowerment of women by including their needs into the developmental goals of the nation, providing them with equal and diverse opportunities for education and participation in the nation's labor force and ensuring improvements in the quality of their lives (Supreme Council of Women 2015). The contribution of women in every field in is not desired or encouraged by the government, but women themselves also assume their role as valued contributors to society. Volunteering during this crisis was no exception.

## 4. Method

We used a mixed qualitative method with a descriptive research design for the purpose of this study. The mixed qualitative method refers to integrating various techniques within a single study qualitatively; for example, interviews are at times combined with document analysis and observations during research (Silverman 2020). We used a combination of focus groups and interviews for this research. The usage of this multimethod approach has several advantages. The first merit is that it helps to strengthen the quality of research—for example, in the epistemological field—as the different methods help to enhance the visibility of numerous perspectives and distinctions.

Additionally, applying more than one qualitative methodological approach provides knowledge that is otherwise inaccessible to the researcher. This is proven by some scholars who refer to research designs that incorporate multiple research strategies to be the strongest

ones (Esterberg 2017). Further, the scholars insist on this merit through their explanation that the integration of interviews with focus groups with photos and diaries plays a massive role in providing a more comprehensive picture of the evaluated phenomenon (Kitzinger 2016). Another advantage of the merging of different qualitative methods is that they provide the researchers with a richer explanation, deeper understanding and analysis results that are better and unbiased (Moran-Ellis et al. 2020). This, in return, increases the reliability of the data among the users and increases individuals' trust in the findings of the study (Teachman and Gibson 2018).

We first conducted a focus group with eight women participants (Table 1). The focus group helped us in assessing responses to volunteering during the pandemic among participants who shared similar backgrounds and demographic elements (Acocella 2019). Although not time-consuming (Byers and Wilcox 2021), the focus group helped us to identify the various motivations of these young women to take up the act of volunteering during a time when most people considered it safer to stay indoors and avoid contact with other people. Full engagement of the participants is another factor that is crucial in any research for accurate and reliable results (Morgan 2019).

The focus group was followed by in-depth interviews with 10 women and 5 male volunteers (Table 1). Both focus groups and in-depth interviews were structured to understand the primary motivations behind volunteering during the pandemic, the volunteering experience and how such experiences influenced the participants' lives. In-depth interviews are a qualitative data collection method that entails direct, face-to-face or telephone engagement. One of the benefits of this technique is that it has none of the many potential dynamics of distractions or peer pressure that threaten focus groups (Boyce and Neale 2017). Although both the focus group and the in-depth interviews were conducted while ensuring the complete confidentiality and anonymity of the participants, the latter provided an opportunity for a one-to-one interaction with the respondent. Minimum or no bias in comparison with focus groups is another added advantage of in-depth interviews (Acocella 2019). This is because the function of the interviewer is often less crucial in discussions, and thus the expected bias is reduced. In addition, there is fairness in the time allocated to each of the respondents, unlike in the focus groups, where the speaking time taken by some of the attendees may be approximately higher compared with others, thus making their input potentially disproportionate (Silverman 2020). Each interview lasted around 50–60 min and was in some cases extended by another 10 or 15 min.

The participants were selected using the snowball sampling technique (Dragan and Isaic-Maniu 2013). The first volunteer was a university student who was referred by a colleague. She helped us in getting in touch with other volunteers that she had worked with during her volunteering activity. Thereafter, the volunteers helped in contacting their fellow volunteers to ask them to be part of the study. Female volunteers were not easy to contact directly but felt more comfortable when a fellow volunteer first spoke to them, and hence the snowballing technique worked better in this case. The participants in the interviews and focus groups were different, to ensure a broad range of viewpoints and insights in the data. Although the participants were contacted through already recruited research participants, their participation was completely voluntary. Before obtaining their consent, they were briefed on the purpose of the research and made aware of their roles and rights as participants in this research. While conducting the focus group, they were addressed via codes (Table 1) to ensure the confidentiality of the participants.

**Table 1.** Participant profiles.

| Code | Gender | Age | Education | Background | Family |
|---|---|---|---|---|---|
| **Focus Group** | | | | | |
| P_1 | Female | 22 | Graduate | Planning for post-graduation | Single. Lives with parents. |
| P_2 | Female | 24 | Graduate | Part-time job | Married. One child. |
| P_3 | Female | 26 | Graduate | Part-time job | Married. Two children. |
| P_4 | Female | 23 | Pursuing graduation | Student | Single. Lives with parents. |
| P_5 | Female | 22 | Pursuing graduation | Student | Single. Lives with aunt. |
| P_6 | Female | 25 | Graduate | Homemaker | Married. Lives with her husband. |
| P_7 | Female | 27 | Post-graduate | Works as a teacher | Married. Lives with her husband and child. |
| P_8 | Female | 23 | Pursuing graduation | Student | Single. Lives with her mother. |
| **Interviews** | | | | | |
| I_w1 | Female | 29 | Graduate | Part-timer | Single. Lives with parents and one sibling. |
| I_w2 | Female | 27 | Graduate | Homemaker | Married. Lives with her husband. |
| I_w3 | Female | 28 | Graduate | Works in her family business | Single. Lives with her parents and two siblings. |
| I_w4 | Female | 22 | Pursuing graduation | Student | Single. Lives with her parents and five other family members. |
| I_w5 | Female | 29 | Graduate | Employed | Single. Lives with parents and three other family members. |
| I_w6 | Female | 26 | Graduate | Not employed | Single. Lives with parents and two other family members. |
| I_w7 | Female | 22 | Pursuing graduation | Student | Single. Lives with parents and one sibling. |
| I_w8 | Female | 25 | Pursuing graduation | Student | Married. Lives with husband and two children. |
| I_w9 | Female | 24 | Pursuing graduation | Student | Single. Lives with parents. |
| I_w10 | Female | 25 | Graduate | Part-timer | Single. Lives with parents. |
| I_m1 | Male | 29 | Post-graduate | Part-timer | Single. Lives with parents and two other family members. |
| I_m2 | Male | 26 | Graduate | Not employed | Single. Lives with parents and three other family members. |
| I_m3 | Male | 25 | Graduate | Employed | Single. Lives with parents. |
| I_m4 | Male | 29 | Graduate | Runs his own business | Married. Lives with spouse and one child. |
| I_m5 | Male | 24 | Pursuing graduation | Student | Single. Lives with parents and two siblings. |

Data were analyzed using thematic analysis, which is consistent with prior studies, to generate themes that characterized the narratives put forward by the study participants

([Braun and Clarke 2006](#)). Braun and Clarke have outlined a series of phases that researchers must carry out to produce a thematic analysis. This necessitated the transcription of the interviews and focus groups, followed by reading and re-reading of those transcripts to identify potential themes. Broad themes emerged from the analysis of the transcripts and were codified. The second level of analysis involved further examination of these codes to identify all-encompassing higher-level themes.

## 5. Findings

The findings of our research helped us to understand the motivations of young women in Bahrain who were volunteering during the pandemic crisis in RT-PCR test drives and vaccination drives.

### 5.1. Normative Factors

The greatest motivation for these young women to take up volunteering during the time of such a severe global crisis was to answer the call of their homeland. Love for the nation and fulfilment of one's duty towards the country was the greatest driver. Many of them received calls from the Ministry of Health asking if they were interested in volunteering or came across a call to volunteer on television.

*"I felt my country was weak at this time and it needed me at this hour of crisis. Something ominous was destroying my nation and I had to be there for my country to save it".* (P_1)

*"I am proud of what I did for my country. My family encouraged me. The volunteering experience developed my social personality".* (I_w4)

Many of them made personal sacrifices to become volunteers. Many volunteers stayed away from their families and stopped socializing during the volunteering period for fear of contracting and passing the infection onto them. For the sake of safety and out of concern for others, they kept themselves isolated. Their love for their country meant that they put the nation before themselves and their families.

*"I stayed away from old parents, extended family members for close to 9 months while I had taken up the volunteering work. I did not socialize at all during that period. We had the best health practices and protocols at the centers and all volunteers underwent regular RT-PCR testing but still I did not want to take any chance. I have young children and I just saw them and my husband during this period. It was very tough to stay away from loved ones, but this was a small price to pay for the opportunity to serve the nation and answer the call of my homeland".* (I_w8)

Serving humanity and philanthropy was another huge motivation to volunteer during the pandemic. Most women who we interviewed or who participated in the focus group told us that witnessing the wide-scale suffering and death caused them a great deal of trauma. Their concern for their families and friends increased and, as each day passed, they felt that the same could happen to their parents or to them or to their children. The desire to do something for humanity and for their fellow human beings was also an outcome of the fact that, during a crisis, it is important look out for our fellow human beings. Only then will someone else look out for us and for our loved ones.

*"I was witnessing a tremor in the world that was altering the course of all our lives. I heard everyday news of widescale suffering and rising death toll. Something inside of me changed and I felt that I needed to do something for the sake of humanity. I could no longer watch the scared faces and helpless eyes".* (I_w3)

Most of them faced initial opposition from their families, whether this was their parents or their husbands, but seeing their resoluteness and service orientation, their families supported them in their initiative. Some of them also told us that they were sitting idle at home, and this was an opportunity to engage in something worthwhile and make a difference to the country, even though it may be in a small way.

*"My parents initially opposed my idea of volunteering during the pandemic. I had to convince them to a certain extent but when I told them that it affected all of us and if everyone thought like this then we will never be able to emerge out of this crisis. Someone had to take a stand and becoming a volunteer was the only way I could help my people as well as myself and my family".* (P_4)

The influence of religion and religious beliefs can also be seen as a motivation for volunteering during the crisis. People are deeply religious in this part of the world. This is even evident in their everyday greetings and expressions, which invariably include the name of the Almighty. Most volunteers told us that their trust in the Almighty and their religious belief of putting others before themselves, sacrificing oneself in their duty to society and in the service of the Almighty, also motivated them to take up volunteering during this crisis.

*"My religion has taught me that my life would remain incomplete if I do not commit myself in some way to the service of humanity. And here I had a great opportunity to not be of service to humanity but to be of use at the time when it was most needed. I knew this was the will of the almighty that I volunteer".* (I_w1)

*"Charitable volunteering, I believe, has its reward that it will help in shielding harm from the virus as much as possible. I trust God, he is the preserver".* (P_7)

Moreover, they also had a deep-rooted belief that if they volunteered and performed good deeds for others, they would be blessed by the Almighty, who would reward them by protecting them and their loved ones from the virus.

*5.2. Personal Factors*

It is interesting how personal suffering and loss can, at times, awaken us to the reality that we need to stop worrying for ourselves and take action to overcome our sense of hopelessness and helplessness. Many of the respondents faced death in the family or of someone close to them due to COVID-19. The trauma and sorrow made them feel weak, lose hope and become helpless and scared. Volunteering was an effort to channel these fears and overcome them.

*"When I and my family were infected with Corona, I saw my family suffer and we lost a dear member of our family. This suffering and loss gave me a motive to apply for volunteering, to work hand-in-hand with other volunteers and frontline workers to fight and eliminate this pandemic".* (I_w5)

*"I lost both my grandparents in a span of 20 days. I could no longer just sit and watch my family suffer. I had to do something. Volunteering gave me strength and hope. Watching me fight join the nation's fight against corona, my family also felt strong".* (P_6)

*"We were socially isolated. Communication between our family and loved ones was cut off. This thing affected me a lot. I woke up every day hoping that the nightmare would disappear. Then I heard from my mother about the volunteering link, and she encouraged me to register for the same. For the first time I had a great feeling, I felt happy, strong, and proud. I wanted to do this till this pandemic was over".* (P_8)

Taking up volunteering made the participants feel empowered, made them feel in control once again and gave them confidence that they could fight this pandemic and emerge victorious. For them, serving others, which helped them to fight the pandemic, was a way to psychologically compensate for their personal losses and tragedies, a way to redeem hope.

All volunteers were very proud of what they had done for the nation and their people. There was a sense of self-actualization in this act, and doing something that made them feel worthy and appear worthy in front of others was definitely an additional motivation for volunteering.

*"I wanted to do something that I can be proud of in the future, and I can tell my kids and later perhaps to my grandchildren that inspires them as well. After taking up a*

*volunteering role, for the first time in my life I felt I was doing something worthwhile".* (P_3)

Many of the volunteers told us that when they decided to take up the act of volunteering, their friends and neighbors asked them to stop visiting them for fear of contracting the virus.

*"I cannot blame them. I think they were perhaps justified in thinking the way they did but that did not deter me for taking up the volunteering role. But the same people later came to me and apologized for their behavior. In fact, they told me that they were proud of me".* (I_w2)

The volunteering act during this crisis became so powerful that it changed social perceptions and made these volunteers national heroes.

### 5.3. Gender Differences in Volunteering Motivation

Another interesting finding of this study was a desire to ensure that women contributed in equal measure as men in taking up volunteering activities. There was an influence of role models, as well as a peer group influence, on some of the women volunteers in deciding to take up volunteering, with evidence of a subtle attempt among some of the women volunteers to prove parity in gender contributions in serving the nation during the crisis.

*"My brother is a doctor and hence a part of the frontline workers fighting corona. I saw him working for 18 h a day without getting tired or lazy. I saw him full of determination and strength. I never hesitated for once when this volunteering opportunity came to me. I wanted to struggle just like my brother, and I wanted to make a mark in my life that I will always cherish".* (I_w4)

*"My friend and I encouraged each other to volunteer for the country. Initially, I did not register on the volunteering link as I feared for the safety of myself and my family, but when I watched other volunteers at the exhibition center either on television or on newspaper, I was excited and motivated to join as volunteer. We also heard that they had a shortage of volunteers at the exhibition center and I and my friend went to volunteer, and thank God, they chose us".* (I_w6)

This is not to say that this was done with any bitterness to prove one gender equal to the other. Instead, it was done with a sense of pride and a drive to ensure that women did not remain behind in standing up with the nation and the population at a time of grave crisis.

We also interviewed a few male volunteers to understand whether the motivations for taking up volunteering during the crisis were similar or if they had different motivations. Male volunteers made similar sacrifices when taking up volunteering work and had the same motivations for volunteering, with the exception of a desire to ascertain equal gender contributions, as seen in female volunteers.

However, there was an additional motivator in the case of male volunteers. For many of them, the motivation to take up volunteering during this period was initially driven by career needs. Volunteering was considered as something that would help them to find a job later, since they would not only gain some experience in supporting the medical services but this experience would help boost their curriculum vitaes and their chances of being employed favorably. Having said this, they also told us that, towards the end of the volunteering experience, they saw it more as an effort to contribute positively towards the common good, rather than as a way to strengthen their career paths.

*"I will be honest, I registered as a volunteer as initially I thought it will benefit me at work or in finding a better job. But now I have changed my thinking and I think more about others and the common good that my work may bring about".* (I_m2)

*"A very short period of volunteering has now become two months of volunteering. I was not very convinced of volunteering in the beginning, but I started volunteering on the*

*advice of a relative that it will benefit me in the future, in terms of work. I continued to work for the people who said beautiful words and gave a beautiful positive energy to me. It made me continue and love what I did. I am now very proud of what I have done".* (I_m4)

### 5.4. Influence of Volunteering Experience on Volunteers

It was also interesting to note in our research how the volunteering experience had a positive effect on the lives of the volunteers themselves. All volunteers recalled their volunteering experience fondly and there was an unmistakable excitement in their tone while narrating their volunteering experiences. The volunteering experience made them feel strong, while becoming useful to their nation and their fellow citizens. The feeling of "one team" was very strong among the volunteers, who expressed sentiments of "one for all and all for one". The training that they received, before being stationed as volunteers, was also appreciated by them. The training focused not only on the "hard skills" required of them to help in RT-PCR testing and support vaccination drives, but also focused on the development of their "soft skills" to help them in performing effective people management:

*"After we were chosen as volunteers, we were given training on collecting swabs for RT-PCR tests, handling samples, managing computers and entering data. But most importantly our center director trained us on the virtue of patience and remaining calm as volunteers. He told us that there may be instances when people come and shout at us, but even then, we must not shout back at them or lose our temper. We must empathize that the ones who are coming to get a test may be infected, may be feeling sick or may have some relative in the hospital or might have lost someone near and dear one to Corona. They might get irritated if they have to wait for long because of heavy traffic at the centers on certain days, but we must maintain our calm. It was not easy at the beginning but gradually I started empathizing more and became better in dealing with people".* (I_w7)

Overall, volunteering was found to have two primary effects on the volunteers.

The volunteering experience helped in their personal growth. Most volunteers told us that the volunteering experience made them more confident, emotionally resilient, instilled a sense of self-worth and self-esteem in them and made them more patient and calmer.

*"Each day during the volunteering period was a challenge. Some days we were very happy and in high spirits. Then on other days, watching others suffer some of us cried in private and it was emotionally overwhelming at times. If one of our teammates contracted the virus, it did not deter us. We took the required tests and followed the health and safety protocols and were back next day for the volunteering work. None of us got scared. We developed a sense of team spirit and that gave us a lot of confidence".* (P_5)

The volunteering experience had, in many ways, influenced as well as altered the course of their future lives and their career choices. Most young women who worked as volunteers now either wanted to work in the social sector or wanted to work for companies who were socially responsible and socially driven. Some of them also told us that they would like to work for the Ministry of Health and continue serving their nation and its citizens. The volunteering experience had given them a purpose in life and a drive for the future.

*"Initially, I had gone to volunteer only for a week. I thought I will be there only for a week and would then get out of there. I was thinking about my old parents. But the spirit of the team and the vision of all the volunteers, their eagerness and determination to eradicate this disease made me stay".* (I_w9)

*"I want to work for the Ministry of Health. I want to dedicate my life in serving people and making a difference to my country through my work. Hopefully, I will have this opportunity in the future".* (I_w10)

## 6. Discussion

Traditionally, reasons for action are classified into "motivating reasons", which explain why a person does something, and "normative reasons", which explain why a person should or should not do something (McNaughton and Rawling 2018).

The motivators behind volunteering for RT-PCR test and vaccination drives identified through this research can also be divided into normative motivators and individual motivators. Motivators such as the call of the homeland, philanthropic zeal, religious values and women as contributors were more normative motivators, which explain why these young women felt that they should take up volunteering during the pandemic; meanwhile, the other two motivators—mitigating personal loss and suffering and self-actualization—were more individual motivators, which explained why these women chose to take up the act of volunteering (Figure 1).

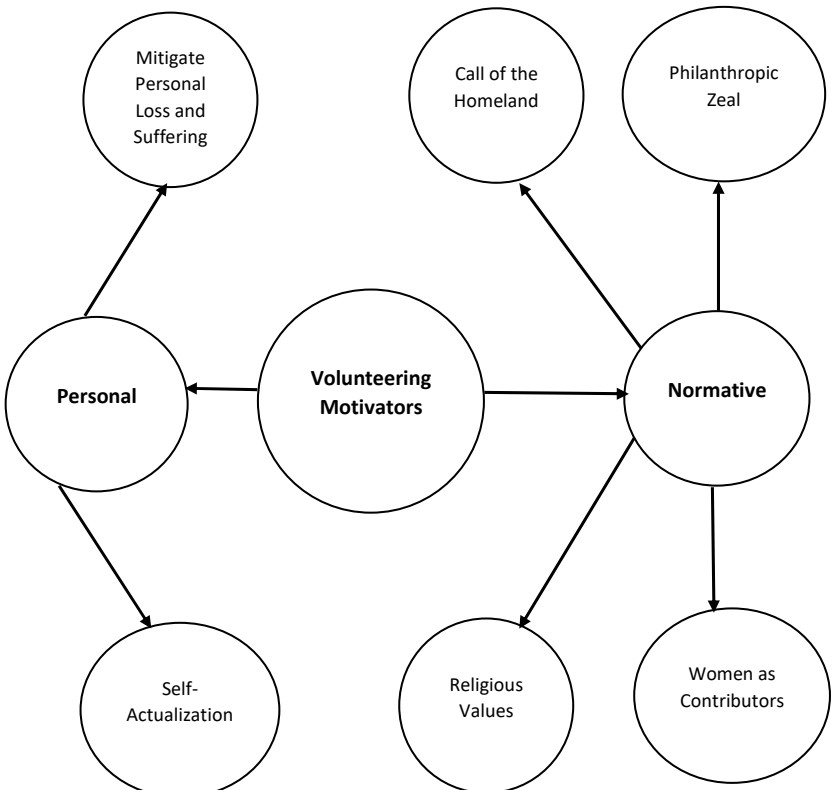

**Figure 1.** Normative and personal motivators of volunteering.

Bahrain is a collectivist society and the influence of the collectivist culture on the motivators behind volunteering among these young women during the pandemic was evident, especially regarding the normative motivators. The normative motivators, answering the call of the homeland, philanthropic zeal and a desire to alleviate the suffering of the community, have a collectivist culture influence. Hofstede defines "national culture" as a "collective programming of the mind which distinguishes the members of one group or category of people from another" (Hofstede and Hofstede 1991). Research also indicates that the national culture may influence the psychology and behavior of the population (Dovidio et al. 2020).

According to Hofstede, values constitute the deepest layer of a culture, and the most significant differences between national cultures are found to be at the level of values (Hofstede et al. 2010). People in collectivist cultures are more concerned about the goals, needs and interests of the group rather than of their own, and vice versa for individualistic cultures, where people give more precedence to their own goals, needs and interests, instead of thinking about the group (Trubisky et al. 1991). The women in this study took up

volunteering because they loved their nation, wanting to do something for their country in the face of a crisis; their love for their community drove them to try to reduce the suffering of their people through their act of volunteering, an act that placed the welfare of the nation and community before their own.

Research evidence shows that people in collectivist cultures are more likely to wear masks (Chu et al. 2020; Lu et al. 2021), fulfil their civic responsibility and express solidarity with their community by following the health mandates and adhering to preventative measures (Maaravi et al. 2021) than those living in individualistic cultures (Stewart 2020).

People in collectivist cultures feel that they have an obligation to support the group. They place the benefits of the group above their individual benefits and, therefore, seek to promote the greater good for the group through their actions and behaviors. The "we" always comes before "me" (Wood et al. 2021). The motivations behind the young women volunteering in Bahrain also appear to be associated with a sense of obligation, a desire to place the interests and benefits of the community and nation before their own and a willingness to sacrifice self-interest for the greater common good.

Unlike the western individualistic culture, Bahrain is a collectivist society and it is the norm for people to interfere in the affairs of others and judge them based on societal standards, including women. Hence, the perceptions of women about themselves is, in many ways, influenced by what society expects from them (Alsaqer 2018). The normative motivator, women as contributors, seems to be influenced by the manner in which the collectivist society expects women to behave and make their contributions. It was their way of fulfilling their obligation towards a nation and society that has provided them with an equal footing in various walks of life.

The cultural identity of Bahrainis stems from the Islamic culture, since close to 70.2 percent of the population is composed of Muslims (Ministry of Information Affairs 2018). In Islamic culture, the core of the societal system is the community and not the individual. This is best explained by the Islamic notion of "Ummah", which considers the whole community as one body, and if any part of the body is in pain, then the whole body is in a state of suffering. Ummah, or the unified Muslim community, points towards collectivism and not individualism (Al Saqer 2016, p. 4). Additionally, in Islamic culture, the idea of a community means that there is no strong difference between public and private and, therefore, whatever is expected of the community at large is also expected from the individual. This is consistent with our finding of religious values being one of the normative motivations behind volunteering. These values drive beliefs that if one's community is suffering, then they are suffering as well, and religion expects them to volunteer during such a time so that, through their actions, they can, in some way, help to ease this suffering.

The literature inarguably shows that millennials are social-minded and volunteering for social causes illustrates a global citizenship attitude (Woosnam et al. 2019). Millennials in more individualistic cultures such as Italy and the United States were motivated to volunteer to reduce their material guilt by giving back to society (Marta et al. 2006) or to improve their career prospects by obtaining credits in colleges/universities, making their resume more impressionable to recruiters (McGlone et al. 2011). Meanwhile, in Israel, which has a stronger collectivist orientation, millennials volunteer as they are driven by their religious values. As a country with a predominantly Jewish population, the Bible has an influence on volunteering, with one of the ten commandments being "Love thy neighbor as yourself", which underlines the significance of helping needy members of society (Kulik 2007). In another collectivist culture, India, passion to serve and personal satisfaction from volunteering were the greatest motivators behind the desire to volunteer (Sengupta 2017). Although the willingness and desire to volunteer is common among millennials, those living in collectivist cultures seem to lean more towards religion and self-actualization as motivators for volunteering.

In our findings, besides religion, we found two personal motivators for volunteering. Among them was self-actualization. Volunteering was a way for participants to boost their self-esteem and self-worth. Volunteering not only made them feel good about themselves

but gave them an experience that they would be proud to tell their children and possibly their grandchildren about.

Volunteering also seemed to influence the participants' career choices and future life choices. Many of them wished to work in the social sector, with their country's Ministry of Health, to continue serving people. Others wished to work only for social-minded organizations. This has significant implications for how companies can attract, recruit and retain these millennial Bahraini women, in a country where women constitute 49% of the total workforce—more than the global average (BNA 2021b).

The success of the government in Bahrain in motivating and mobilizing volunteers and in dealing with the challenge of the pandemic also holds important lessons for other nations, an example worth emulating for not only tackling future pandemics, but also in reprioritizing certain critical UN sustainable development goals to lead socio-economic recovery post-pandemic (Nair et al. 2021).

## 7. Conclusions

Our research helped us to understand the motivations behind young women in Bahrain accepting volunteering opportunities during the pandemic crisis. There were both normative as well as personal motivators (Table 2). The normative motivators were those that indicated why the women felt that they should be volunteering during this crisis to support government-sponsored RT-PCR test drives and vaccination drives. The personal motivators were the individual reasons why the women took up volunteering during the pandemic. We also compared and contrasted the motivations of these women with those of male volunteers, and while most of the volunteering motivations remained common between both genders, the women had an additional motivator in ensuring that gender contributions remained equal, while male volunteers also had the motivation of boosting their career prospects by volunteering, although the thought of working for a common good never eluded them while they were engaged in the act of volunteering.

**Table 2.** Motivating reasons and motivators of women volunteers.

| Motivating Reasons | Motivator |
| --- | --- |
| Call of Homeland | Normative Motivator |
| Philanthropic Zeal | Normative Motivator |
| Religious Values | Normative Motivator |
| Women as contributors | Normative Motivator |
| Mitigate Personal Loss and Suffering | Personal Motivator |
| Self-Actualization | Personal Motivator |

We also found that culture had a huge influence in shaping these motivations. Collectivist cultures such as Bahrain place the common good and the interests of the group before individual interests and even sacrifice the latter to meet the former. The influence of culture is clearly evident, with most of the motivators behind volunteering among these women falling under the normative category. Religion, although part of the culture, played an explicit role in shaping their motivation for volunteering. The generational cohort and identified values of millennials in this case, and the fact that the women chosen for this study were all from this cohort, also had a role in shaping their motivations.

The findings assume importance not only in understanding what will motivate young women towards volunteering but also the fact that these experiences had a lasting effect on these women, so much so that it influenced their future careers as well as their development as people. Their growing preference towards working in the social sector means that non-profits have a good opportunity to attract a very active and significant cohort to join their workforce or participate as volunteers. This also is a message for companies, indicating that their social-mindedness and socially responsible behavior would not only help in attracting this important part of the working population to join their workforce but retain them through positive engagement. The findings also hold important lessons for other nations

and their governments in terms of strategies to mobilize volunteers during a national or global crisis.

The findings can be generalized to other collective cultures, particularly Islamic collectivist cultures, when it comes to understanding the motivations of young women in taking up volunteering; however, for non-collectivist cultures, the same set of motivations may not apply. For non-Islamic collectivist cultures, the religious dimension of motivation needs to be reinvestigated with respect to the prevailing religious values in each place. The limitation of this research also serves as a cue for further research.

In the future, research can be conducted to investigate volunteer motivations in various cultural contexts that have not been covered in this research, e.g., individualistic cultures and collectivist non-Islamic cultures, focusing on both women and men. Studying volunteering behavior is interesting because it holds many important lessons for businesses regarding how to attract and engage their primary stakeholders (employees and customers) and counter the challenges posed by the pandemic. Non-profits who experience difficulty in attracting young talent to work for various social causes in often inhospitable locations with unattractive remuneration and perks might also benefit from understanding such motivations to re-strategize their talent management.

**Author Contributions:** Conceptualization, D.S.; Methodology, D.S.; software, D.A.-K.; validation, D.S., D.A.-K.; formal analysis, D.S.; investigation, D.S.; resources, D.A.-K.; data curation, D.S., D.A.-K.; writing—original draft preparation, D.S., D.A.-K.; writing—review and editing, D.S.; visualization, D.S.; supervision, D.S., D.A.-K.; project administration, D.S.; funding acquisition, No external funding. All authors have read and agreed to the published version of the manuscript.

**Funding:** This research received no external funding.

**Institutional Review Board Statement:** Ethical review and approval were waived for this study as it did not involve any particular institutions and individual volunteers participated voluntarily based on their individual capacity and consent. Further, all ethical research practices were followed that protected the rights of the participants of this research.

**Informed Consent Statement:** Informed consent was obtained from all subjects involved in the study.

**Conflicts of Interest:** The authors declare no conflict of interest.

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
