# Peer review of "Motivations of Young Women Volunteers during COVID-19: A Qualitative Inquiry in Bahrain"

_admsci, doi:10.3390/admsci12020065_

Round 1

Reviewer 1 Report

This is an interesting piece of work and I learned so much about the link between volunteering and its context. There are few comments to improve the paper so it could be publishable:

1) Method: what criterion you used to select the participants? is there any difference between those participated in the focus group and in-depth interviews? There is no explanation on how data was analysed, please explain how the data analysis lead to your Figure 1. 

2) Findings: sub-headings could be provided to differentiate the normative and personal factors. There are a number of quotes but who said that and the person's background plus how (interviews/focus group) could be helpful. A profile table on those participated in the interviews and focus group could be added. 

3) The language style is ok but there are multiple errors throughout the manuscript, please get a professional copy-editor to check. For example, line 79, 142, 157, 224 (fight against), 228-229 (confusing sentence), 246-249 (just a quote without any link to your argument), 260 (to delete 'in'), etc. 

Reviewer 2 Report

Dear authors,

I had the opportunity to read your work and it seemed very interesting to me. It is contextualized in a theme of great current and interest for lectures, and for society in general. I also consider that the method used is interesting, as the interviews allow us to bring to light numerous aspects that would otherwise not be possible.

I consider that the introduction perfectly addresses the study problem. The objective is also clearly exposed. I understand that the last sentences of the introduction (67-73 lines) recognize the contribution of the paper. If not, I would tell you about to introduce some idea that expresses this contribution.

The literature review is also good. The theme and the various aspects related to it are exposed with a wide range and sufficient references. You must correct a mistake that appears on line 79. Therefore, I consider that the theoretical framework is good.

Regarding the analysis of the context, it is also correct. I have the same opinion of the method section. It is adequate and provides a lot of information. However, I would like to suggest the possibility that you carry out some table that covers the main ideas; also contributions from authors on the different aspects that are noted from the interviews; that is to say, instead of putting the paragraph nothing more, to support these ideas in previous authors who have dealt with the theme. The set of impacts, both positive and negative, could also be collected in a table, breaking them down by theme, and also if they are personal or normative. Furthermore, a DAFO analysis could be carried out, which I believe would complement this section very well. So, you would not see only the text, but it would be more complete with these tables that I suggested.

As for the conclusions: you must expand the limitations. You talk about them, but they are scarce. Perhaps you can mention aspects such as the size of the sample, the possibility of complementing this qualitative analysis with a quantitative one, etc.

As for the references, I consider that they are adequate and up to date.

Much success.

Reviewer 3 Report

Dear authors,

It was pleasure and veru big interest in reading your paper.

Here are some comments for the improvement of your manuscript:

1) Introduction. This part misses explanation why the specific country as Bahrain was selected for the case study. This is very important to reveal even very shortly the content of this country's specific in volunteering, gender issues, etc. already in this part (not only in the chapter 3). The scientific novelty of the research should be clarified too. The concrete scientific problem (problematic questions) should be identified too. The aim and objectives of the study provided in more structured way would be be very helpful to understand the structure of the whole manuscript too. Besides, the method of the research could be mentioned too. This part should be supplemented and improved with mentioned issues.

2) Literature review. I guess some technical issue appread in the fiest paragraph of this part "Error! Bookmark not defined."

"Volunteering" and "Volunteering Prominent Features" parts are prepared quite goo, but it seems that the newest resourse, used for this part comes from 2012. I would highly suggest to make supplements to this part using some most new and relevant studies (from 2019–2022).

"Volunteering During Crisis" is based on resources form 201-2014. But we know that pandemic as the crisis started in 2020. So authors should use more relevant resources to base their insights in this chapter.

3) Method. There are some places, where it is hard to get what is just explained as theoretical possibilities and what was done in the real research. "We used a combination of focus group and interviews for this research. For instance, combining interviews with document analysis and observations during research (Silverman, 2020)" - so, is the first sentence describes what was done, or both are true? As the second one looks like just a theoretical mentioning, but not realized.  So you are recommended to review this part. I would recommend to provide the scheme of research design or a table with columns (stage, method, sample, instrument, organizing, ethics) - which could visualize the whole view of the empirical research methodology.  I miss information about sampling criterias (why such samples, how interviewees and participants of focus group were selected (criteria)) and research instruments (the structure of questionaires (questions) for focus groups and interviews, thematics, the theoretical backkground for them, etc.).

Authors mention, that "Although both focus group as well as in-depth interviews were conduced ensuring complete confidentiality and anonymity of the participants...". Confidentiality was ensured by not providing personal details to authors of each quotation. But I don't want to agree that authors ensured the anonimity. Did you as researchers know with whom you spoke in focus groups and interviews? If yes, then you did not ensured the anonimity (this is mostly possible in quantitative research without direct meeting of researcher as respondent). If you did not know to whom you speak, then how did you ensured that your participants are really those to whom you should speak?  So, we - readers need more clarification, how the sample was collected and if really you ensured anonimity for your participants.

4) Results. I would highly suggest to provide quotes from interviews and focus groups provided in Italic and with codes, which let readers to identify, by which one of method this quote was got and which interviewee or participant provided it. This is very important yoto see and know how much of mentioned samples were important to this part (maybe just two of interviews are presented here as the reader does not know which line belongs to whom). Codification helps to keep the confidentiality, but still ensure transparency of results' presentation.

Codification of answers (citations from interviews) could be like: I_w1 – I_w10 and I_m1 – I_m5, where "I" - means "Interviewee", "w" - woman, "m" - man, 1 – 10 - the number of interview according to the order of interviews in the research.

For example,

"<citation....> (I_w6)"; "<citation....> (I_w8); "<citation....> (I_m2)"

Codification of answers (citations from the focus group) could be like: P_1 – P_8, where "P" - means "Participant", 1 – 8- the number of participant, provided by the researcher.

For example,

"<citation....> (P_1)"; "<citation....> (P_3); "<citation....> (P_5)"

I would like to see a table with the summarized and provided list of motivators (motivating reasons) of women volunteers at the end of this chapter.  In the table they should be connected to motivators, identified in the discussion part (Figure 1). It would help to see better strcuture of your reserch results and to see links.

5) Discussion. There is no reference to the Figure 1 in the text as "(see Fugure 1)", so it is not clear when readers should pay attention on it. Besides, it is not clear what is the source for this figure - authors conducted, based to some resources or taken from somewhere?

6) Conclusions. Despite of interesting insights I did not find even one paragraph (here and in discussion) which could explain how identified motivators are different (the same) for women volunteers during different periods (pandemic or before it) - it has to be compared with implications from this one research with researches, made before pandemics. The specifics of COVID-19 must be really seen and emphasized by authors as the emphasis on it is in the title of the paper.

Authors did not provided limitations of their research  in details. It must be supplemented.

Future research directions avoid any idea about the comparison between pandemic and post-pandemic results, even it could be very interesting issue.

7) Authors do not use the template of the journal fully, so they check this and do amendments:

  • References in the text (not APA style - please check);
  • List of references (not in the alphabetic order, not in the APA style) - must be corrected;
  • Contribution of authors - not identified still.

Good luck in improving your paper.

Round 2

Reviewer 3 Report

All needed corrections according to my previous comments were supplemented. The paper was improved and now provided good structure and content.

Good luck.